The effect of repeated laser stimuli to ink-marked skin on skin temperature—recommendations for a safe experimental protocol in humans

Madden Victoria J. 1
Catley Mark J. 1
Grabherr Luzia 1
Mazzola Francesca 1
Shohag Mohammad 1
Moseley G. Lorimer 1 2 lorimer.moseley@gmail.com
Sansom Institute for Health Research, University of South Australia , Adelaide, South Australia , Australia
Neuroscience Research Australia , Sydney , Australia
Tobin Desmond
Electronic publication date: 2016 Jan 14
Publication date: 2016
Volume: 4
Electronic Location ID: e1577
Received 2015 Oct 2; Accepted 2015 Dec 16
Copyright: ©2016 Madden et al.
Copyright year: 2016
Copyright holder: Madden et al.
License: This is an open access article distributed under the terms of the Creative Commons Attribution License, which permits unrestricted use, distribution, reproduction and adaptation in any medium and for any purpose provided that it is properly attributed. For attribution, the original author(s), title, publication source (PeerJ) and either DOI or URL of the article must be cited.
License URL: https://creativecommons.org/licenses/by/4.0/

Keywords: Lasers, Safety, Forearm, Skin temperature, Back, Skin, Pain, Nociception, Evoked potentials

Funding: Pain, Mind and Movement SIG of the IASP South African Society of Physiotherapy Oppenheimer Memorial Trust, South Africa Australian Postgraduate Award scholarship Swiss National Science Foundation Australian National Health & Medical Research Council Principal Research Fellowship 1061279 VJM received scholarships from the Pain, Mind and Movement SIG of the IASP and from the South African Society of Physiotherapy to present part of this work in poster form at the 15th World Congress on Pain. VJM is supported by the Oppenheimer Memorial Trust, South Africa. MJC is supported by an Australian Postgraduate Award scholarship, LG is supported by the Swiss National Science Foundation, GLM is supported by an Australian National Health & Medical Research Council Principal Research Fellowship (ID 1061279). The funders had no role in study design, data collection and analysis, decision to publish, or preparation of the manuscript.

==============================
Background. Nd:YAP laser is widely used to investigate the nociceptive and pain systems, generating perpetual and laser-evoked neurophysiological responses. A major procedural concern for the use of Nd:YAP laser stimuli in experimental research is the risk of skin damage. The absorption of Nd:YAP laser stimuli is greater in darker skin, or in pale skin that has been darkened with ink, prompting some ethics boards to refuse approval to experimenters wishing to track stimulus location by marking the skin with ink. Some research questions, however, require laser stimuli to be delivered at particular locations or within particular zones, a requirement that is very difficult to achieve if marking the skin is not possible. We thoroughly searched the literature for experimental evidence and protocol recommendations for safe delivery of Nd:YAP laser stimuli over marked skin, but found nothing.

Methods. We designed an experimental protocol to define safe parameters for the use of Nd:YAP laser stimuli over skin that has been marked with black dots, and used thermal imaging to assess the safety of the procedure at the forearm and the back.

Results. Using thermal imaging and repeated laser stimulation to ink-marked skin, we demonstrated that skin temperature did not increase progressively across the course of the experiment, and that the small change in temperature seen at the forearm was reversed during the rest periods between blocks. Furthermore, no participant experienced skin damage due to the procedure.

Conclusion. This protocol offers parameters for safe, confident and effective experimentation using repeated Nd:YAP laser on skin marked with ink, thus paving the way for investigations that depend on it.

Introduction

Laser stimulation is an important tool for investigating the nociceptive and pain systems, because it allows for the selective activation of nociceptive neurons with a brief, tightly controlled stimulus. Although CO2 lasers have been widely used, the skin discolouration they produce (Lefaucheur et al., 2012) and difficulties with using them in small spaces and at awkward angles have prompted a shift in favour of solid state lasers (e.g., Cruccu et al., 2003). Solid state lasers such as Neodymium Yttrium–Aluminium–Perovskite (Nd:YAP) lasers may be preferable due to their greater flexibility: the machines tend to be less cumbersome, and transmission of the laser through a flexible fibre-optic cable means that these stimuli can be delivered inside an MRI tunnel (Perchet et al., 2008). Many experimental protocols require repeated stimulus application (for example, to measure thresholds, assess acuity, etc.). However, when laser is used for repetitive stimulation there is a risk of progressive temperature increase if the same skin area is heated too often. Progressive increases in skin temperature over the course of a procedure may (a) elevate the risk of skin damage and (b) compromise the reliability of the outcome being studied—for example, perceptual acuity, perceptual intensity or ERP amplitude. This risk may be elevated in people with chronic pain because there are emerging data that suggest spatially defined dysfunction in thermoregulation and cortical stimulus processing in these groups (e.g., Moseley, Gallace & Iannetti, 2012; Moseley, Gallace & Spence, 2012; Moseley, Gallagher & Gallace, 2012). To reduce this risk, the location of the stimulus is usually shifted slightly between trials to prevent heating of the same skin area in successive trials. This shifting of location is also thought to reduce sensitisation or habituation to the stimulus (Iannetti et al., 2003; Wiech et al., 2010).

This risk of temperature increase and consequent skin damage presents an obvious problem because, to our knowledge, there are no published data outlining the number or frequency of Nd:YAP laser stimuli that is safe. A further issue is the quandary presented when researchers want to accurately localise stimulation sites: it is generally considered that marking the skin with ink is problematic because darkened skin absorbs the radiant heat more rapidly, exposing it to risk of damage (Leandri et al., 2006). One option would be to mark the skin with white ink. However, pale markings are difficult to see on pale skin, making this approach unsuitable for procedures in which an experimenter must quickly localise stimulation sites. Furthermore, marking stimulation sites with a colour that is difficult to see makes experimenter fatigue more likely, and could thus compromise accuracy. Using black pens for skin markings is common practice in other experiments that deliver stimuli to human skin and would be a convenient and sensible option for studies that use laser stimuli, if such markings can be used without increasing skin temperature enough to compromise reliability or cause skin damage.

An additional ‘safety’ check that is commonly used in studies involving painful stimuli is to calibrate the stimulus intensity to individual participants according to pain rating (Mancini et al., 2011; Wager, Matre & Casey, 2006; Weiss et al., 1997). This is a reasonable approach for preventing undue suffering that could be caused by the experimental pain percept. However, pain is influenced by a multitude of factors, including attention (Villemure & Bushnell, 2002), salience (Wiech et al., 2010), emotion (Wiech & Tracey, 2009), task demands (Petrovic et al., 2000) and expectations (Atlas & Wager, 2012). Consequently, stimulus calibration according to pain percept may not be a useful strategy to ensure safety. The finding that people with chronic pain have alterations in perceptual acuity (Wand et al., 2011) suggests that the stimulus energy-percept mismatch may be even more pronounced in people with pain, emphasising that experimental procedures for studying pain must be tested for safety in those with and those without pain.

Here we present data from an experimental protocol in which multiple stimuli were applied to the black-ink-marked skin of 15 people with chronic low back pain and 13 healthy control participants, and thermal imaging was used to evaluate skin temperature before and after stimulation blocks. We quantified the effect of repeated stimulation on skin temperature to ascertain the safety of an inter-stimulus interval and block-rest-block protocol when the locations of stimuli have been marked with black ink.

Methods

Participants

We recruited adult volunteers using flyers and word of mouth. Volunteers were not eligible if they had sensation problems, diagnosed peripheral vascular disease, diabetes mellitus, or psychological or neurological problems, or unusual skin conditions (e.g., dermographism) that might compromise skin safety with laser application. Volunteers with skin too dark for possible erythema to be noticed were also ineligible. Participants had to be pain-free or have chronic back pain, defined as pain between the levels of T12 and S1 (with or without concurrent leg pain) (Merskey & Bogduk, 1994). Volunteers with back pain were excluded if they had concurrent neck or arm pain, were pregnant or less than six months post-partum, or had had spinal surgery. Participants were remunerated at AU$20/h for their involvement. All procedures conformed to the Helsinki Declaration and were approved by the institutional ethics committee.

Experimental procedure

Participants received laser stimuli to the skin of the forearm and the back. Prior to stimulation, the hair on the dorsal forearm and the back was trimmed using clippers, and a template was used to draw two dot-grids onto the skin with a black Artline® 200 fine 0.4 pen. Dots were spaced 4 mm apart in the proximal-distal (forearm) or medial-lateral (back) direction, and 5 mm apart in the radial-ulnar (forearm) or cephalo-caudad (back) direction. The forearm grid measured 48 mm × 20 mm; the back grid measured 80 mm × 20 mm (see Fig. 1).

Figure 1 Dot-grids as drawn onto forearm (A) and back (B).

Two dot-grids are pictured at each site: the second was used for tactile stimulation (data not presented here).

Stimuli

Laser stimuli were delivered using an Nd:YAP laser (Deka: Stimul 1,340, wavelength 1,340 nm, spot diameter measured at 3.5 mm, pulse width 6 ms), which has previously been shown to activate Aδ nociceptors specifically (Iannetti, Zambreanu & Tracey, 2006). The intensity of laser stimulus for each participant was established, on the basis of an ascending staircase procedure, before testing began. The operator delivered laser stimuli of changing intensity, asking participants to describe what they felt. Once a painful pinprick was elicited, participants rated the pain intensity on a 0–10 numerical rating scale with anchors of “no pain” (0) and “worst pain ever felt” (10). The operator aimed to identify the intensity that consistently elicited pinprick pain of 2–3 out of 10, and this was used for the testing procedure. If a participant reported a diminution or an increase in pain intensity during the procedure, the operator adjusted the laser stimulus intensity accordingly, with a maximum permitted laser intensity of 2.0 J. This level was established during piloting as a limit that would prevent skin damage, and thus became an additional exclusion criterion (if a laser stimulus intensity of 2.0 J produced less than 2/10 pinprick pain). Participants received stimuli in pairs. Three blocks of 14 stimulus pairs were delivered to the forearm and three blocks of 16 pairs to the back, with at least 30 s between pairs and 3–8 min between blocks (see Fig. 2). The interval between pairs was selected to reduce perceptual habituation, and the break between blocks was chosen to allow participants relief from the sustained concentration required by the perceptual acuity task. Blocks were randomly ordered. Each stimulus was delivered over a dot in the dot-grid. The two stimuli of each pair were delivered 4–44 mm (at the forearm) or 4–76 mm (at the back) apart. This approach was part of a wider plan to evaluate perceptual acuity at each location. It was impossible for any one location to be stimulated twice without a break of at least 120 s. The operator monitored skin colour visually, so as to detect any localised erythema that could indicate undue skin heating.

Figure 2 Diagram depicting experimental procedure and main results.

CLBP: people with chronic low back pain.

Outcome

We used infrared thermal imaging (FLIR SC620 camera, FLIR systems, Oregon, USA; sensitivity < 40 mK, field of view = 24 × 18°) to record the average skin temperatures within the demarcated zones before and after each stimulation block. This camera produced an image that is colour-coded by infrared radiation. The ThermaCAM Researcher Professional software (version 2.9, FLIR systems, Oregon, USA) allows the user to delineate a certain area and calculate absolute temperature parameters for that area. In this study, the demarcated stimulation zone was delineated and the calculation was made as an average for that area. Skin condition was visually monitored throughout the procedure.

Statistical analysis

Temperature data were analysed using two analyses of variance (ANOVA): (1) temperatures before and after each block were compared using repeated-measures ANOVA with Time (before/after block) and Site (forearm/back) and Block (1/2/3) as within-subject factors, and Group (patient/healthy) as the between-subject factor, and (2) forearm skin temperatures before each block were compared using repeated-measures ANOVA with Block (1/2/3) as the only factor. Planned comparisons were used to investigate significant effects, and Bonferroni adjustments were applied to correct for multiple comparisons. Alpha was set at 0.05. Where the assumption of sphericity was violated, adjusted values are reported, with degrees of freedom also adjusted accordingly.

Results

In this experiment, each participant received 84 stimuli at the forearm and 96 stimuli at the back, delivered over a period of ∾70 min.

We recruited 29 adult volunteers, one of whom was excluded for dermographism. Of the 28 adults tested, 15 of them had chronic low back pain (mean age ± SD = 43 ± 17 years; 8 female) and 13 were pain-free (mean age ± SD = 49 ± 14 years; 8 female).

Overall, skin temperature was greater at the back than at the forearm (main effect of Site, F(1, 26) = 9.23, p = .005). Skin temperature increased over each block (main effect of Time, F(1, 26) = 19.45, p < .001) and the extent of the increase was greater in the forearm than it was in the back (Time × Site interaction, F(1, 26) = 30.028, p < .001). Comparisons of the Time × Site interaction showed that forearm skin temperature increased over a block (p < .001, mean change ± SE = 0.40 ± 0.07 °C, 95% CI [0.27–0.54]) and that back skin temperature did not change over a block (p = .098, mean change ± SD = −0.08 ± 0.05 °C, 95% CI [−0.02–0.18]). There was no main effect of Group (p = 0.938) or Block (p = 0.626) on skin temperature and there were no other significant interactions.

Despite the small change in forearm temperature over each block, there was no difference between skin temperatures at the start of each block (no effect of Block, p = .248), indicating that the rest periods between blocks were sufficient for the forearm skin temperature to return to baseline.

None of our participants had localised erythema sufficient to warrant termination of the procedure, none had any visible indications of skin damage on completion of the experiment, and none reported adverse effects afterwards.

Discussion

In this study, Nd:YAP laser stimuli were applied to the ink-marked skin of 28 participants without resultant skin damage or lasting heating of the skin. No participant reported skin damage due to the procedure. The thermal imaging data showed no overall increase in skin temperature across the course of the experiment. Skin temperature at the back did not change within blocks. Skin temperature at the forearm increased within each block, but that increase was reversed during the rest periods between blocks such that skin temperatures at the start of each forearm block did not differ.

These results demonstrate that it is possible to use Nd:YAP laser stimuli repeatedly over ink-marked skin without inducing skin damage, provided that strict safety parameters are observed.

There were four factors that ensured the safety of this design. The first factor was strict exclusion criteria. We excluded participants with neurological or sensation problems, and we ensured that skin tone was pale enough for the operator to detect any laser-induced change in skin colour. Although Nd:YAP does not produce the same discoloured skin lesions that CO2 laser is known to, whether it produces lesions of deeper layers of the skin is unknown (Iannetti, Zambreanu & Tracey, 2006). We therefore considered the visual monitoring of skin colour a necessary precaution in this study, and planned to exclude any volunteer in whom this would not be possible.

The second factor was the selection of a conservative upper limit to the intensity of laser to be used in the experiment. This limit was based on pilot work in our lab, during which two participants (both males with Fitzpatrick skin type III, 50 and 24 years old) developed minor skin lesions after repeated application of stimuli over 2 J, despite reporting less than 2/10 pain at such intensities. Those pilot subjects would not have been excluded on the basis of the other exclusion criteria. We therefore recommend a conservative upper limit to the intensity of laser to be used in an experiment, so as to identify and exclude participants who may be at risk of skin burn during a procedure in which the laser stimulus intensity is determined according to subjective ratings. Our results also indicate that calibration of stimulus intensity according to pain report was not dangerous in this group of participants with and without chronic back pain. Considering that people with chronic pain may display a greater energy-percept mismatch than their healthy counterparts (Wand et al., 2011), there is reason to expect that report-based calibration of stimulus intensity may be unsafe. However, this study did not find evidence of skin damage with the stimuli used, which may be attributable to the 2 J safety limit on stimulus energy.

The third factor that ensured participants’ safety was the use of break periods, which allowed for recovery of skin temperature towards a baseline level between stimulation blocks. The fourth factor that ensured the safety of this design was that the ink markings used in this study were small dots made with a 0.4 mm-diameter pen. As such, the blackened skin area constituted a very small fraction of the area over which the laser stimulus was delivered (mark diameter 0.4 mm; beam diameter 3.5 mm), probably resulting in a differential heating effect over the area of surface exposed to the beam. Blackened skin is expected to absorb Nd:YAP laser more superficially, leading to quicker absorption and dissipation of heat, and more selective activation of Aδ fibres than would be expected in un-blackened skin (Leandri et al., 2006). In this study, the rate at which skin temperature recovered from laser stimulation was likely linked to the comparatively small area over which the skin was blackened. Any future work based on this report will need to consider these four factors in order to achieve equivalent safety.

Limitations

This report provides safe parameters for Nd:YAP laser stimulation using a 3.5 mm spot diameter. Further work would be required to determine safe parameters for other spot sizes, because the area over which the stimulus is delivered affects the extent of skin heating. If researchers require moment-by-moment information on skin temperature changes, real-time thermal imaging will be necessary. We used imaging before and after blocks, and are therefore unable to provide data on skin temperature between stimuli within a block.

A reasonable concern about this report is that we did not personally reassess participants’ skin condition in the days following the procedure. It is possible that participants developed delayed signs of skin damage, particularly considering that any strongly absorbent regions of tissue could be subject to greater risk of heat-induced damage due to an unequal heating effect. However, participants were explicitly asked to report any signs of skin damage, and we received no such reports. Furthermore, the intensity limit of 2 J was established on the basis of pilot testing that showed no skin damage in pilot participants, according to visual assessments made immediately after the procedure and in the following days. We are therefore confident that the parameters presented here did not result in visible skin damage in this group of 28 participants.

Conclusion

This procedure offers parameters for safe and effective experimentation using Nd:YAP laser and black ink skin markings when delivering stimuli to the back and the forearm of healthy participants and participants with chronic back pain.

Supplemental Information

Supplemental Information 1 Raw data on skin temperature in all participants

Average temperature data for healthy (denoted 0) and CLBP (denoted 1) at pre-block and post-block testing, for 3 blocks.

Click here for additional data file.

The authors are grateful to Dr Tasha R Stanton for sharing her extensive knowledge about Nd:YAP laser stimuli. Part of this work was presented at the 15th World Congress on Pain (2014).

Additional Information and Declarations

Competing Interests

Author Contributions

Human Ethics

Data Availability

The authors declare there are no competing interests.

Victoria J. Madden conceived and designed the experiments, performed the experiments, analyzed the data, wrote the paper, prepared figures and/or tables, reviewed drafts of the paper.

Mark J. Catley conceived and designed the experiments, performed the experiments, analyzed the data, wrote the paper, reviewed drafts of the paper.

Luzia Grabherr performed the experiments, analyzed the data, wrote the paper, reviewed drafts of the paper.

Francesca Mazzola and Mohammad Shohag performed the experiments, reviewed drafts of the paper.

G. Lorimer Moseley conceived and designed the experiments, contributed reagents/materials/analysis tools, wrote the paper, reviewed drafts of the paper.

The following information was supplied relating to ethical approvals (i.e., approving body and any reference numbers):

Human Research Ethics Committee of the University of South Australia approved protocol number: 30447.

The following information was supplied regarding data availability:

Raw data has been provided in the Supplemental Information.

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
