# Peer review of "The effect of repeated laser stimuli to ink-marked skin on skin temperature—recommendations for a safe experimental protocol in humans"

_PeerJ, doi:10.7717/peerj.1577_

## Round 0.1 · original submission · Major Revisions

I agree with the reviewers' comments, some points have been highlighted by all reviewers e.g., you have used a black colored ink to draw the dots where another lighter color with lower absorption of 1340 nm light (e.g., white) may be been preferable? Also, you will be need to more clearly state the rationale and motivation for this study (e.g., to include two groups with and without pain).

·

Basic reporting

No Comments

Experimental design

1. Why and how did participants receive stimuli in pairs? Were the two stimuli of each pair delivered with a fixed distance?
2. Since blackened skin is known to absorb more radiant energy, do you think it is safer to use an ink color which has less absorption at 1340 nm wavelength?

Validity of the findings

No Comments

Reviewer 2 ·

Basic reporting

The focus is on safety of laser stimuli for diagnostic purposes. The experiments are reported faithfully, though the method section needs some clarification (see comments)

Experimental design

The experimental design is just acceptable. It should have been done more carefully. There are several flaws summarized in the comments section. These are not serious enough as to suggest rejection, but should be given attention

Validity of the findings

What they found is clearly stated and respondent to the aims of the paper. Some data could have been more thoroughly exlpoited.

Additional comments

The study addresses the issue of safety and standardization in the use of Nd:YAP lasers for diagnostic purposes (perception of nociceptive threshold and LEPs) in marked skin. The experiments are fairly sound, but there are a number of questions that should be answered.
1) Why the need to mark the skin?. The authors say to identify definite spots on the skin. However the authors only consider black colour. Did they take into account that with a different colour the problem of undesired absorption could be overcome (white, for example)? Could they please discuss this issue?
2) Skin was marked with a matrix of tiny black spots spaced 5x4mm. So the marked surface was minimal. What is the rationale of such design? Is it because they wanted to avoid any significant increase in temperature at the skin surface and only use marks to define targeted skin areas? If so, they should be aware that blackening the skin makes the laser beam more selective to Adelta receptors, as the rise in temperature is much more superficial. They irradiated a spot of 3.5mm in diameter (they say “size”, please specify better that size means diameter) centred on a mark 0.4mm diam. They were then in a condition of rather uneven stimulation. When irradiating the black spot they were in a similar condition as the CO2 laser (definitely selective for A delta, but prone to burning), whilst when irradiating unmarked skin, heating was deeper and stimulation possibly not as selective. This should be discussed and clarified.
3) How did they synchronize the camera with the stimulus? I guess that the temperature lecture was an average either in space (camera definition is just 640x480 pixels) and in time. But which time? And what was the dimension of the framed picture? Actually, we won’t know the actual temperature reached by the marked spots, where some burning could have taken place. Did they remove the ink after the experiments and checked with a dermatoscope possible damage to the skin at those tiny sites?
4) I couldn’t find the stimulus duration. Of course if the energy of 2J is delivered in a short time, it will cause a steeper rise in temperature than a long time. Please specify.
5) Energy of 2J with (wavelength of 1349nm) delivered to a spot of 3.5mm diam gives an energy of 208mJ per sq mm, which is very high indeed and likely to cause burns even in unmarked skin. Is there anything amiss or wrong in the method section?
6) The authors do not make it clear whether the laser stimulus was delivered with the aim of assessing a perception threshold or for evoked potentials.
7) There is no attempt to correlate the stimulus with perception. Did the authors find any difference between marked and unmarked areas as to perception to a given stimulus? I am aware that this is not within the aim of the paper, but it might be a useful information

·

Basic reporting

The study addresses the problem of inducing skin damage by a Nd:YAP laser during stimulation of nociceptive neurons. To obtain precise data about spatial processing of noxious input marking the skin with a dark ink is commonly applied. However, the ink-marked skin is more susceptible to injury by laser due to increased absorption in the ink.
The authors proposed a protocol consisting of exclusion criteria, an upper limit for pulse laser energy, a relaxation time between stimulation blocks, and small markings compared to irradiation area. By testing the protocol on 28 volunteers, it was demonstrated that the Nd:YAP laser can be used repeatedly over the ink-marked area without inducing skin damage.
An important problem for performing clinical experiments was considered in the present manuscript, i.e. participant/patient safety. This study results could help designing future experiments in the field.
The text is well written and comprehensible.

Experimental design

There are a few comments about the experiment, which should be considered by the authors:
- Delayed effects were not considered or presented. Erythema, hypo /hyperpigmentosis can appear after few hours or days.
- Measured surface temperatures does not necessary correlate with heat deposition/temperature rises in tissues. In case of strong absorbers deeper in a tissue, a significant temperature rise in the absorber area can result in moderate or even low surface temperature rise. For example, experimentally measured temperature depth profiles induced by a Nd:YAP laser in human skin are presented in Milanic et al., Energy deposition profile in human skin upon irradiation with a 1,342 nm Nd:YAP laser, Las. Surg. Med. 45(1), 2013
- From the description in the Methods section it is not clear how the skin temperature rise was measured (ls. 90-92). Was the amplitude or a certain time average measured? Or as mentioned in the Limitations subsection after the stimuli? If the latter is true, was image always taken at the same time interval after the stimuli?
- The skin types, age, and sex of the participants were not reported. The study findings can significantly depend on these parameters.
- The pain levels were mentioned in the Participants subsection (l.56) without providing a reference.
- The authors do not explain why the separations used in the dots pattern were selected (ls. 65-67). Are there any physiological or thermodynamic reasons for the choice?
- The pulse length should be reported in the Stimuli subsection (ls.71-73). The pulse length significantly affects heat deposition in skin.
- The authors state that the upper limit exposure of 2 J was determined by a pilot experiment performed on two volunteers (ls. 80-83). It is important to present important information about the volunteers, e.g. age, sex, and skin type.
- The authors specify the pause lengths between subsequent stimuli and blocks (ls.83-88) without providing a reasonable explanation.

Validity of the findings

The statistical analysis of the reported results was suitable. However, the following comments should be considered by the authors to improve the results and discussion part of the manuscript:
- It is not necessary to repeat the number of participants (ls.105-106).
- A table showing information (skin type, age, sex) and the study results (pulse energy, pain level, side effects, temperature rise, etc.) should be presented.
- A figure showing the most important findings of the study would significantly help readers to understand them.

Additional comments

- What was the motivation for including two groups of volunteers: one with and one without chronic pain? What is the significance of the results obtained for these two groups? Neither the motivation nor the results explanations are clear from the text.
- Authors should explain why dominantly black colour is used to draw dots. Would it be possible to use other markers with lower absorption of 1340 nm light? It is not uncommon in the laser medicine to use white markers which absorb only a small portion of laser pulse energy.

---

## Round 0.2 · accepted · Accept

Thank you for comprehensively taking on board the reviewers' comments and on improving the manuscript accordingly. Specifically, you have more clearly highlighted for the reader some of the key limitations of this study, which cannot be removed via subsequent revision. I am confident this study will make an important contribution to the field.